# Endoscopic Ultrasound-Guided Treatment of Gastric Varices Using Coils and Cyanoacrylate Glue Injections: Results after 1 Year of Experience

**DOI:** 10.3390/jcm8111786

**Published:** 2019-10-25

**Authors:** Sławomir Kozieł, Katarzyna Pawlak, Łukasz Błaszczyk, Mateusz Jagielski, Anna Wiechowska-Kozłowska

**Affiliations:** 1Department of General, Transplant and Liver Surgery, Medical University of Warsaw, 02-097 Warsaw, Poland; endokoziel@gmail.com; 2Department of Gastroenterology, Hospital of the Ministry of Internal Affairs in Szczecin, 70-382 Szczecin, Polandannamwk@wp.pl (A.W.-K.); 3Department of General, Gastroenterological and Oncological Surgery, Ludwik Rydygier Collegium Medicum in Bydgoszcz, Nicolaus Copernicus University in Torun, 87-100 Torun, Poland; matjagiel@gmail.com

**Keywords:** gastric varices, endoscopic ultrasound, coil, cyanoacrylate, obliteration

## Abstract

Background and Aims: Gastric varices (GVs) occur in 20% of patients with portal hypertension. GVs are associated with a 65% risk of bleeding over the course of 2 years and have a mortality rate of up to 20%. The standard treatment for GVs is obliteration with cyanoacrylate (CYA). This study presents our experience with combined therapy (vascular coils and CYA) under endoscopic ultrasound (EUS) guidance. Methods: 16 patients (9 male and 7 female) were included into our study. Etiology of portal hypertension included: portal vein thrombosis (PVT) (31.0%), isolated splenic vein thrombosis (SVT) (25.0%), alcoholic cirrhosis (12.5%), hepatitis C cirrhosis (19.0%), and alcoholic cirrhosis with PVT (12.5%). Varices type GOV-2 were diagnosed in 8 patients, type IGV-1 and IGV-2 in 6 and 2 patients, respectively. Indications for treatment were based on endoscopic and endosonographic evaluations of GVs. Inclusion and exclusion criteria were also specified. Technique depended on the size of varices (different size of coils + CYA additionally). The results were based on the achievement of technical success, therapeutic effects, and number of adverse events. Average follow-up period was 327 days. Results: From January to August 2017, 16 patients were treated with EUS-guided obliteration of GVs using vascular coils only or coils with CYA injections. 6 (37.5%) and 10 (62.5%) patients underwent primary and secondary prophylaxis for hemorrhage, respectively. Technical success was achieved in 15 patients (94.0%). Mean numbers of implanted coils and CYA volume during one procedure were 1.7 and 2 mL, respectively. Therapeutic success was achieved in all patients treated with the combination. There were no serious complications such as embolization or death due to the procedure. Three patients (19.0%) had transient abdominal pain and two (12.5%) had transient fever. 1 patient had clinical symptoms of gastrointestinal bleeding. Conclusions: Based on our retrospective research we have concluded, that EUS-guided implantation of intravascular coils combined with cyanoacrylate injections is an effective method of treatment with an acceptable number of complications.

## 1. Introduction

Gastric varices (GVs) occur in 20% of patients with portal hypertension, mainly due to portal system obstruction and various forms of liver cirrhosis. The incidence of esophageal varices in cirrhotic patients is around 5% at the end of one year, 28% at the end of three years, while small varices progress to large varices at a rate of 10% to 12% annually [1].

Increased portal vein pressure (HVPG; hepatic vein wedge pressure-free hepatic vein pressure) more than 10 mmHg and hepatic venous pressure gradient more than 5 mmHg induce the development of collateral circulation through the left and posterior gastric veins, short gastrointestinal veins, right gastro-mesenteric vein, and veins connecting the submucosal venous plexus of the stomach. These pathological portal-systemic connections cause the development of GVs [2].

There are three main classification of gastric varices: Hashizome, Arkawa’s, and Sarin classification, which is most commonly used. The classification presented by Sarin et al. [2] in 1992, was developed based on the analysis of 568 patients with portal hypertension of various etiologies and with primary and secondary gastric varices. It describes the types of varices in the upper gastrointestinal tract and the incidence, risk of bleeding, and mortality rates [3]. This classification distinguishes two types of GVs: isolated GVs (IGV) and gastroesophageal varices (GOV). Type 1 IGV are fundal varices that present in the cardia in the absence of esophageal varices (IGV1) or that occur in the stomach outside of the cardio-fundal region or the first part of the duodenum (IGV2). GOV that are a continuation of esophageal varices into the lesser curvature are known as type 1 (GOV1). Type 2 (GOV2) are esophageal and fundal varices that continue with the greater curvature of the stomach (Figure 1) [3].

Pathological vessels surrounding the gastrointestinal wall can be precisely visualized during EUS examination. Endosonography allows for evaluation of every type of vessel localized in the neighbor of the stomach and esophagus. Moreover, the use of color Doppler can reliably differentiate gastric varices from other non-vascular lesions [4]. Pathological vessels of submucosal plexuses, entangling the lumen of the gastrointestinal tract, can be most easily found by endoscopic examination. Deeper pathological venous vessels, inaccessible in endoscopic examination, was characterized by Soderlund et al. [5] This classification allows determination of the types of venous vessels based on their location:-periesophageal/perigastric, directly adjacent to the wall of the wall,-paraesophageal/paragastric, non-direct contact with the gastrointestinal wall;-perforating veins, connecting superficial vessels with the para- and peri- vessels (Figure 2).

The most common type of gastric varices is GOV type 1. They account for 74% of all gastric varices, however incidence of bleeding is highest with IGV type 1 [6]. GV are a serious complication of portal hypertension and are responsible for approximately 10% of varicose hemorrhages from the upper gastrointestinal tract. Hemorrhages from GV are often severe, with a high mortality rate of up to 20% [2]. The most important predictors of hemorrhage are location (IGV1 > GOV2 > GOV1), the size of varices > 5 mm, HVPG > 12 mmHg, decompensated cirrhosis (Child B or C), and the endoscopic presence of red wale marks [6]. This rate depends not only on the degree of variceal advancement but also on the availability of highly specialized care and treatment options offered by the center.

The role of endoscopic glue (N-butyl-2-cyanoacrylate) injection in primary prophylaxis for GVs is well established [7]. Mishra et al. [8] presented in their study, that glue injection was associated with lower bleeding and mortality due to GVs, compared to nonselective beta blocker [8]. Wherefore, more effective and safer treatments are being sought. The new additional option may be application of coil into the lumen of GVs. Coils were first used to treat ectopic varices by Levy et al. [9], this technique has been increasingly implemented into clinical practice. CYA injection alone carries a risk of embolization [1]. The coil has synthetic fibers, which constitute the scaffolding for CYA, reducing the risk of embolism. Furthermore, the role of fibers is to slow down the blood flow in the vessel and promote blood-clot formation occluding the vessel [1]. Therefore, the use of an intravascular coil was investigated to determine its potential as a new treatment for gastric varices.

The objective of the study is to report our experience using coils and CYA glue injection in the treatment of gastric varices.

## 2. Methods

The Endoscopic Unit of the Department of General, Transplant, and Liver Surgery is a center that treats patients with esophageal varices and GVs. Every year, we perform more than 700 endoscopic variceal obliterations, and we treat more than 100 patients with fundal varices. The Endoscopic Unit of the Hospital of the MIA in Szczecin is one of the leading centers in Poland that uses EUS for treatment. Inspired by the experiences of our colleagues in California and Spain, we decided to join forces and use this new method to treat varicose veins in our patients [3,10,11].

The study was approved by the bio-ethic commission. Every patient gave written consent and was thoroughly and in detail informed about the course of the study.

All the patients included into research fulfilled the follow criteria:-conglomerate of gastric varices type GOV 2 and IGV 1 located in the gastric area, and varices IGV 2 located in the duodenum with a minimum diameter of 5 mm, visible in the endosonographic image.-in a case of varices’ diameter ranging from 5 to 10 mm, implantation of intravascular coils alone was performed. In a case of varices’ diameter above 10 mm, implantation of intravascular coils with cyanoacrylate injections was performed.

Patients with gastric or duodenal varices of diameter under 5 mm in endosonographic view were excluded from the study. Moreover, patients under 18 years old and pregnant women were also excluded from the study.

16 patients (9 male and 7 female) were included into our study, in age range 29–75 years. Etiology of portal hypertension was differential and included: portal vein thrombosis (PVT) (31.0%), isolated splenic vein thrombosis (SVT) (25.0%), alcoholic cirrhosis (12.5%), hepatitis C cirrhosis (19.0%), and alcoholic cirrhosis with PTC (12.5%). 9 out of 16 patients in our study were diagnosed with gastric varices due to portal thrombosis (Child A—7/9 patients and Child B—2/9 patients). All the other (seven patients) experienced gastric varices due to liver cirrhosis (Child B—6/7 patients and Child C—1/7 patients). None of the patients from the study group underwent TIPS procedure. 3 out of 6 patients from the study group who were diagnosed with liver cirrhosis underwent liver transplants. Varices type GOV-2 were diagnosed in 8 patients, type IGV-1 and IGV-2 in 6 and 2 patients, respectively. The procedure, as a primary prophylaxis, was performed in 37.5% of patients, while as a secondary prophylaxis in 62.5%. All the patients included in our study were treated with nonselective betablockers (propranolol) given in maximal tolerated doses (Table 1).

All procedures were performed during a total of 5 sessions by 2 endoscopic physicians (A.W.K. and S.K.). The sessions were alternatively performed in Warsaw and Szczecin. Indications for treatment were determined after an endoscopic evaluation and meeting all inclusion criteria. The procedures were performed under general anesthesia with endotracheal intubation while the patient was on the back or in the left side position. Before the procedure, all patients received antibiotic prophylaxis (1 g of third-generation cephalosporins, intravenously). The linear echo-endoscope EG-3870-UTK (Pentax^®^ Europe GmbH, Hamburg, Germany) and the US Preirus platform (Hitachi Medical Systems Europe, Zug, Switzerland) were used to administer treatment.

During the first stage of the procedure, varices were evaluated using standard endoscopy. After the examination, the stomach was filled with saline solution and the varices were assessed with EUS. We focused on the following: determining the size of the varices; evaluating the flow rates on color Doppler identifying the perforating vessels supplying the variceal veins; and determining the optimal method of accessing the varices (transesophageal-transcrural or transgastric approach).

To puncture the varicose veins and implant the coils, standard fine-needle aspiration was performed using EchoTip^®^Ultra 19G needles (Cook Endoscopy, Limerick, Ireland). After puncturing the varicose veins, a needle stylet was used to push the implanted 0.035-inch embolic coils. We used two types of coils: Tornado^®^ coils, which were 5 mm or 10 mm in diameter and 12.5 cm in length, and Nester^®^ Embolization Coils, which were between 5 and 18 mm in diameter and 7 to 14 cm in length (Cook Endoscopy) (Figure 3). Using the same needle, CYA glue (Histoacryl^®^; B. Braun, Melsulgen, Germany) diluted in Lipiodol (Lipiodol^®^ Ultrafluid; Guerbet, Aulnay-sous-Bois, France) in a 1:1 (0.5 mL + 0.5 mL) ratio was injected in the lumen of the vessel. After the procedure, hemostasis was checked and vascular flow on color Doppler was evaluated. Technical success was achieved when coil implantation was completed, whereas therapeutic success was defined by the lack of active bleeding and ligation of vascular flow in the obliterated varicose veins.

During follow-up, patients were assessed endoscopically and endosonographically at 1, 3, and 6 months, (Figure 4). All 16 patients presented for the first check-up as out patients. 13 patients (81.0%) underwent additional elective procedures. The average follow-up period was 327 days (range, 182–443 days). Therapeutic success was achieved in all 12 patients treated with coil implantation and CYA (75.0%). However, for 4 patients in whom CYA was not injected, active blood flow in the treated varices was diagnosed using EUS during the first check-up. This required repeated obliteration using coils and CYA. 1 patient required re-implantation of 2 more coils, 9 months after the first procedure. 5 patients who presented for subsequent examinations required obliteration of residual varices with CYA. 7 patients underwent esophageal variceal ligation due to esophageal varices. No serious complications were observed after the procedure. 3 patients (19.0%) had transient abdominal pain and 2 (12.5%) had fever persisting up to 5 days after the procedure. Symptoms subsided after oral administration of non-steroidal anti-inflammatory drugs. 1 patient was hospitalized in a peripheral center due to gastrointestinal bleeding symptoms (dark stool, hemoglobin decreased by 2 g %, no hemodynamic disturbances). After endoscopy was performed, no active bleeding site was found, and the patient did not require any further endoscopic treatment. The procedure did not result in complications such as emboli or deaths. During the observation period, three patients with liver cirrhosis dropped out of the study due to elective liver transplantation.

## 3. Results

From January to August 2017, 16 patients were treated with obliteration of GVs using vascular coils (4 patients) or with coils and EUS-guided CYA injections (12 patients). A total of 21 coil implantations were performed. There were 7 female patients and 9 male patients with an average age of 51 years (range, 29 to 75 years).

Patient data are summarized in Table 1 and Table 2. All procedures were elective. For 6 patients (37.5%), the procedures were performed as primary prophylaxis. However, for 10 patients (62.5%), the procedures were performed as part of secondary prophylaxis against hemorrhage. 6 patients previously underwent esophageal variceal ligation, and 4 patients previously underwent obliteration using CYA. For 9 patients, the cause of portal hypertension was portal vein thrombosis or isolated splenic vein thrombosis (56.0% in total). For 7 patients (44.0%), portal hypertension was caused by liver cirrhosis. Two cases (12.5%) of portal vein thrombosis occurred simultaneously with alcoholic liver disease-dependent cirrhosis. 8 patients had GOV2 diagnosed, 6 had IGV1 diagnosed, and 2 patients presented IGV2 located in the duodenum. 1 patient from the GOV2 group developed additional varices in the duodenum, which were the cause of bleeding. The average size of the varicose veins was 17 mm (range, 5 to 45 mm). In 12 cases, enlarged perforating veins were visible.

Technical success was achieved in 15 out of 16 patients (94.0%). For 1 patient with fundal and duodenal varices, the duodenal varix ruptured during the implantation of coils, which caused massive bleeding. The hemorrhage was managed with CYA injections.

The number of implanted coils during one procedure ranged from 1 to 3 (average, 1.7), and the volume of the administered CYA (0.5 mL Histoacryl + 0.5 mL Lipiodol) ranged from 1 mL to 9 mL (average, 2 mL).

## 4. Discussion

Hemorrhages from varices of the stomach often have a dramatic course and usually require treatment in an intensive care unit. The risk of bleeding depends on the size and location of the varices, and it increases with the duration of the disease. Approximately 70% of GVs are GOV1, but they are responsible for only 11% of bleeding cases. However, IGV1, which comprise 8% of GV, are responsible for approximately 80% of hemorrhages [12]. The risk of bleeding is approximately 65% during a two-year period, and bleeding recurrence after spontaneous hemostasis occurs in 35–90% of patients [13,14]. It should be remembered that isolated varices of the stomach often arise in patients with normal liver function and result from portal vein thrombosis or locoregional portal hypertension, which may be caused by thrombosis or anomalies of the splenic vein. Therefore, after managing the bleeding, the prognosis is better for these patients than for those with liver disease or poor liver function [15].

According to the recommendations of the Baveno VI Consensus, the method of choice for endoscopic treatment of bleeding GVs is obliteration with CYA glue. However, there are no recommendations for the use of these adhesives for the prevention of hemorrhage [16]. Nevertheless, their use as primary and secondary prophylaxis for hemorrhage seems to be a good option, especially for large GOV2 or IGV1 with signs of increased bleeding risks [6,7,8,10,11,12,13,14,15,16,17,18,19,20,21]. Depending on the size and extent of the varices, patients may require multiple CYA injections during repeated treatments. However, these are associated with an increased risk of emboli complications. Intravascular coil implantation induces a clotting process in the varices, and the coil is a type of scaffold for polymerizing the adhesives. This reduces the risk of blockages and reduces the amount of injectable CYA that is required. A standard endoscopic examination is the gold standard for assessing esophageal and fundal varices. However, it only allows the evaluation of superficial varices that impinge the lumen of the stomach. The use of EUS makes it possible to identify expanded vessels, peri-gastric vessels, and perforating vessels, which are the direct cause of variceal development [4,13,14,22,23]. EUS also allows for the precise placement of a coil into the varix or perforating vessel, thus closing its blood supply. In addition, assessing the blood flow on color Doppler allows for a direct assessment of the effectiveness of the procedure.

In comparison with other studies, we used transesophageal and intra-gastric access to puncture the varices [3,10,24]. Transesophageal access provides a stable setting for the echoendoscope and facilitates the puncture. However, it only provides good access to varices around the cardia. Varices located in other parts of the stomach require gastrointestinal access. This can result in difficulty puncturing the varices due to the lower stability of the endoscope and the laxity of the varices. Therefore, it is important to use very sharp needles. In our opinion, the needles described here meet these requirements. The diameter of the puncture needle (19G) determines the thickness of the coils to be inserted (0.035 inch) and the type of CYA adhesive. To extend the polymerization time, we used N-butyl-2 CYA, which requires dilution with Lipiodol. It is a thick, fatty substance that is difficult to inject during a short time period through a smaller-diameter needle. The 2-octyl CYA used by the authors in San Francisco (California) is not available in Poland.

The effectiveness of combined therapy (coils plus CYA) does not differ from that presented in the literature [3,10,11,24]. Contrary, Lobo et al. [25] evaluated in randomized controlled trial (32 patients), safety and efficacy of EUS-guided coil plus cyanoacrylate (group I) versus conventional cyanoacrylate technique (group II), in the treatment of gastric varices. In both groups, the majority of patients required only one single session for varix obliteration. They also noticed that both techniques have similar efficacy in the obliteration of varices. However, in other studies, significantly less CYA adhesive was needed to obliteration the varices, compared to procedures with CYA alone (2 mL vs. 7 mL) [11,15,25,26,27,28]. Furthermore, poor long-term efficacy has been observed in patients who have been treated only with coils (without adhesives). Despite the closed blood flow through the lumen of these varices observed with EUS immediately following the procedure, partial recanalization was observed in the majority of patients during the first follow-up examination. These patients required re-implantation of the coil and CYA injections. Lobo et al. [25] analyzed that both techniques have similar efficacy in the obliteration of varices. It was result of the small sample size in this study [25].

We noted more complications than Bhat et al. [12] (31.5% vs. 7.0%). However, these included minor complications (transient pain and fever) and one case of bleeding that did not require endoscopic re-intervention [12]. Our group did not experience any major complications such as embolism or casualties. In turn, Lobo et al. [25] reported 18.8% of epigastric pain as an early complication, and 25.0% as a late complication [25]. Pulmonary embolism was observed in 4 (25.0%) patients [25]. Also, a greater tendency towards embolism was observed in the group treated using CYA alone [25].

## 5. Conclusions

Based on our retrospective research we have concluded, that EUS-guided implantation of intravascular coils combined with cyanoacrylate injections is an effective method of treatment with an acceptable number of complications.

## Figures and Tables

**Figure 1 jcm-08-01786-f001:**
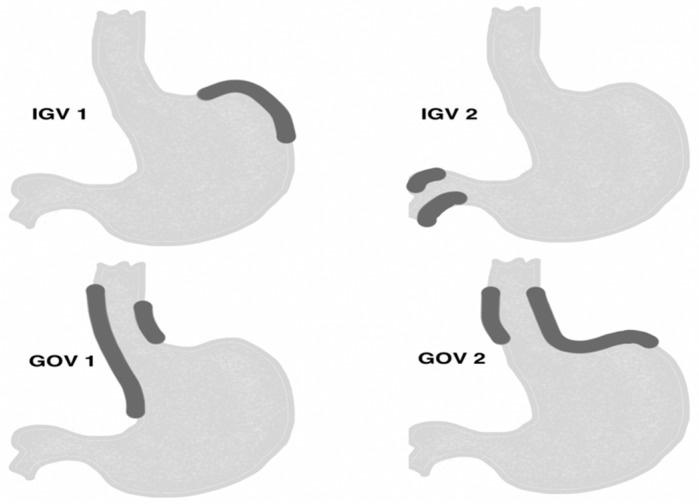
Sarin’s classification [2].

**Figure 2 jcm-08-01786-f002:**
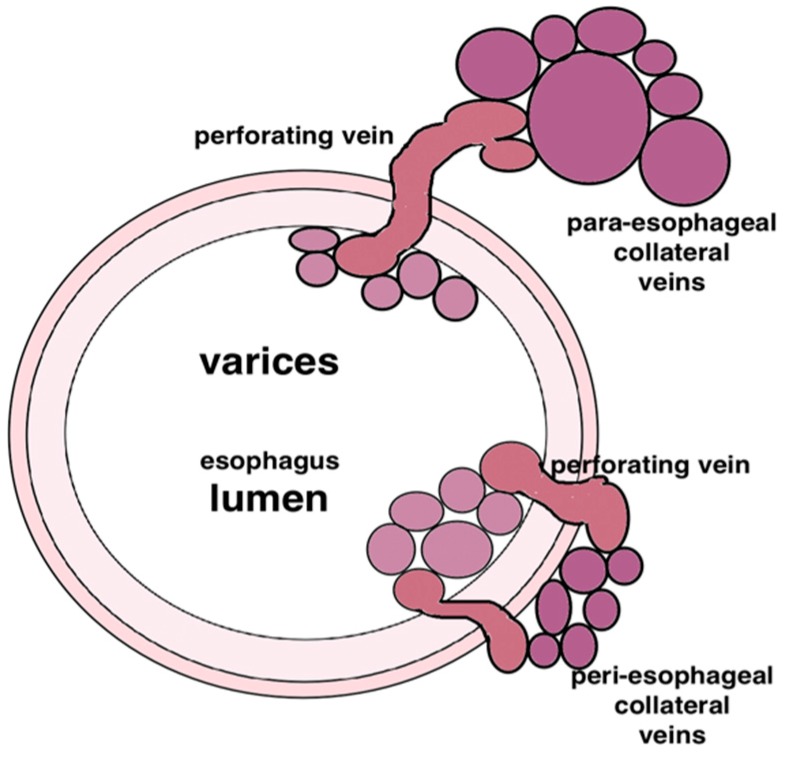
Types of pathological vessels connected with portal hypertension. Soderlund et al. [4].

**Figure 3 jcm-08-01786-f003:**
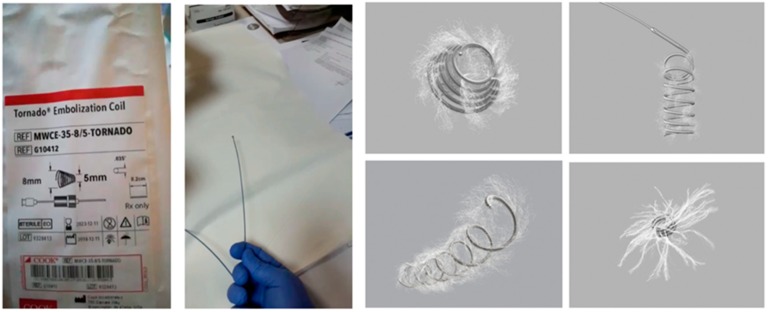
Types of coils.

**Figure 4 jcm-08-01786-f004:**
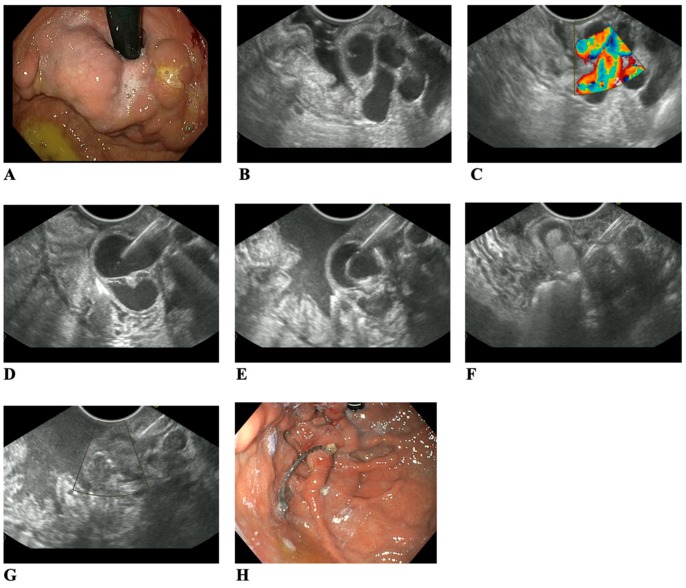
(**A**) Large gastric varices type IGV-1; (**B**) EUS view of variceal conglomerat; (**C**) Turbulent flow in color Doppler; (**D**) Varix puncture by 19G needle; (**E**) Coil placement through a needle; (**F**) CYA glue injection through the same needle; (**G**) Absent flow in color Doppler few minutes after procedure; (**H**) Endoscopic view of excreted coil and scar after eradicated varices, 3 months follow-up. EUS, endoscopic ultrasound; CYA, cyanoacrylate; IGV, isolated GVs.

**Table 1 jcm-08-01786-t001:** Characteristics of the study patients.

Parameter	Type	Value	
Total number of patients	Male	9 (56.0%)	Total: 16
	Female	7 (44.0%)	
Mean age (range)	51 (29–75)		
Portal hypertension etiology	Portal vein thrombosis (PVT)	5 (31.0%)	
	Isolated splenic vein thrombosis (SVT)	4 (25.0%)	
	Alcoholic cirrhosis	2 (12.5%)	
	Hepatitis C cirrhosis	3 (19.0%)	
	Alcoholic cirrhosis + PVT	2 (12.5%)	
Varices type	GOV-2	8 (50.0%)	
	IGV-1	6 (37.5%)	
	IGV-2	2 (12.5%)	
History of bleeding	Primary prophylaxis (never bled)	6 (35.0%)	
	Secondary prophylaxis (recent hemorrhage)	10 (65.0%)	
Gastric varices	due to portal thrombosis	Child A—7 (77.8%) Child B—2 (22.2%)	Total: 9
	due to liver cirrhosis	Child B—6 (85.7%) Child C—1 (14.3%)	Total: 7
Nonselective betablockers (propranolol)	given in maximal tolerated doses		

**Table 2 jcm-08-01786-t002:** Results of the treatment, adverse events, follow-up.

Parameter	Type	Value	
Previous varices treatment	EVL	6	
	CYA glue injections	4	
Mean varices size in mm (range)		17 mm (5–45)	
Method of treatment	Coils + CYA	12 (75.0%)	
	Coils alone	4 (25.0%)	
Mean coil number (range)	1, 7 (1–3)		
Mean CYA glue/Lipiodol mixture volume in mL (range)	2 (1–9)		
Technical success	15 (94.0%)		
Follow-up data (month; number of patients)	First follow-up (1–3 month)	16 (100%)	
	Second follow-up (3, 6, 12 month)	13 (81.0%)	
	Mean follow-up in days (range)	327 (182–443)	
Therapeutic success	After first procedure	12 (75.0%)	
	Coils + CYA patients (*n* = 12)	11 (92.0%)	
	Coils alone patients (*n* = 4)	0 (0%)	
	After 2nd and 3rd (*n* = 5)	5 (100%)	
Adverse events	Pain	3 (19.0%)	Total: 6 (37.5%)
	Fever	2 (12.5%)	
	Minor bleeding	1 (6.0%)	
	Severe (embolization, serious hemorrhage, death)	0 (0%)	

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
