# Peer review of "Endoscopic Ultrasound-Guided Treatment of Gastric Varices Using Coils and Cyanoacrylate Glue Injections: Results after 1 Year of Experience"

_jcm, 2019, doi:10.3390/jcm8111786_

Round 1

Reviewer 1 Report

This is an interesting firs experience paper, looking at the EUS based coil- and glue based obliteration of gastro-esophageal varices.

There are more than 40 publications on the subject, and this therapy is currently being considered as one of the techniques of choice for endoscopic management of the problem.

There are few concerns/suggestions on the paper:

Informed consent - this is retrospective human study and it has to have informed consent/IRB approval - this must be stated in the article As the novelty of the paper is limited, the article should be more educational, e.g. provision of Table of classification-schematic-picture classification of esophageal varices would add to the value of the paper. Technical description of the procedure should be improved, and made more step-based, with illustrations accompanying each of the steps Data on the follow-up methods are not presented in the methods section, rather, in the results section There is no comparison of cyanoacrylate alone with cyanoacrylate+coils; adding some background data on this would improve the scientific value of the paper

Author Response

Szczecin, 13 October 2019

Dear Editor,

Thank you very much for thorough and critical revision of our manuscript entitled “Endoscopic ultrasound-guided treatment of gastric varices using coils and cyanoacrylate glue injections: Results after 1 year of experience”. We have made the appropriate corrections in the manuscript. We would also like to respond to all the Reviewers’ remarks in this letter.

Our answers to the Reviewers’ remarks:

Reviewer #1:

Major comments

This is an interesting first experience paper, looking at the EUS based coil- and glue based obliteration of gastro-esophageal varices. There are more than 40 publications on the subject, and this therapy is currently being considered as one of the techniques of choice for endoscopic management of the problem.

There are few concerns/suggestions on the paper:

Informed consent - this is retrospective human study and it has to have informed consent/IRB approval - this must be stated in the article.

Every patient signed a written consent and was thoroughly and in details informed about the course of the study. Considering the retrospective character of our study the local bio-ethic commission gave its consent to carry out our research.  

As the novelty of the paper is limited, the article should be more educational, e.g. provision of Table of classification-schematic-picture classification of esophageal varices would add to the value of the paper. Technical description of the procedure should be improved, and made more step-based, with illustrations accompanying each of the steps.

We are very grateful for the comments. We agree with the Reviewer that inclusion of information mentioned above (table of classification-schematic-picture classification of esophageal varices and technical description of the procedure step-by-step) will increase educational and essential value of our publication. 

Data on the follow-up methods are not presented in the methods section, rather, in the results section.

According to the reviewer’s recommendations the data on the follow-up methods are now presented in the Methods section.

There is no comparison of cyanoacrylate alone with cyanoacrylate+coils; adding some background data on this would improve the scientific value of the paper

We agree with the Reviewer’s comments, but comparison of cyanoacrylate alone, coils alone and cyanoacrylate with coils is a subject of our other prospective study. This study is in progress at the moment and its results will be presented in our next publication. The research will involve much larger study group of patients with gastric varices treated with used endoscopic treatment. 

We hope that made corrections will satisfy both the Reviewers and the Editors.

Kind regards

Katarzyna Monika Pawlak

Reviewer 2 Report

The authors of the manuscript entitled "Endoscopic ultrasound-guided treatment of gastric  varices using coils and cyanoacrylate glue injections: Results after 1 year of experience" report their experience on the treatment of gastric varices with EUS. They included 16 patients over a period of 8 months, achieving high rate of technical success.  The results suggest that the use of coil + cyanoacrylate is safe and effective. 

After reading the paper, I have some comments and questions.

ABSTRACT:

Line 21. The design of this study does not allow to conclude which treatment is more effective. A randomized clinical trial will be required to obtain this type of conclusion. In this type of design the conclusion should be related to the results obtained regarding technical success, rate of rebleeding and complications.

INTRODUCTION:

Line 44: "This study presents the results of the first study of obliteration of GV with vascular coils and 45 injections of CYA glue under the guidance of endoscopic ultrasound (EUS) in Poland".  The authors should change the objective of the study, to something more specific such as " The objective of the study is to report our experience using  coils and CYA glue injection in the treatment of gastric varices".

METHODS:

The authors should follow the STROBE recommendations for this type of manuscript

Specify the design of the study.

Specify the criteria of inclusion and exclusion of patients

Can the authors include information about why some patients were treated with coils and CYA and other patients only with CYA. Was there a change in the protocol of management of these type of patients?

Does the rate of rebleeding was zero during the follow up? In the manuscript  was only reported 1 episode of bleeding without known source. In other publications similar to this one the rate of rebleeding was reported around 20% of cases.

How many patients were under treatment with betablockers?

What was the stage of the disease in case of cirrhotic patients that were included?

Were other treatments like TIPS offered to patients included after the treatment with coils.

DISCUSSION

The study " SAFETY AND EFFICACY OF EUS-GUIDED COIL PLUS CYANOACRYLATE VERSUS CONVENTIONAL CYANOACRYLATE TECHNIQUE IN THE TREATMENT OF GASTRIC VARICES: A RANDOMIZED CONTROLLED TRIAL" supports the findings of the study. It would be interesting to include this study in the discussion

Have the authors any explanation for the high prevalence of adverse outcomes reported in the study in comparison to to other studies?. Was it something related with the experience of the endosonographers in the technique?.

Author Response

Szczecin, 13 October 2019

Dear Editor,

Thank you very much for thorough and critical revision of our manuscript entitled “Endoscopic ultrasound-guided treatment of gastric varices using coils and cyanoacrylate glue injections: Results after 1 year of experience”. We have made the appropriate corrections in the manuscript. We would also like to respond to all the Reviewers’ remarks in this letter.

Our answers to the Reviewers’ remarks:

Reviewer #2:

Major comments

The authors of the manuscript entitled "Endoscopic ultrasound-guided treatment of gastric varices using coils and cyanoacrylate glue injections: Results after 1 year of experience" report their experience on the treatment of gastric varices with EUS. They included 16 patients over a period of 8 months, achieving high rate of technical success.  The results suggest that the use of coil + cyanoacrylate is safe and effective. 

After reading the paper, I have some comments and questions.

ABSTRACT:

Line 21. The design of this study does not allow to conclude which treatment is more effective. A randomized clinical trial will be required to obtain this type of conclusion. In this type of design, the conclusion should be related to the results obtained regarding technical success, rate of rebleeding and complications.

We fixed miswriting. We agree with the argument the design of the study does not allow to reach a conclusion concerning the effectiveness of both treatments. Based on our retrospective research we have concluded, that EUS-guided implantation of intravascular coils combined with cyanoacrylate injections is an effective method of treatment with an acceptable number of complications. That conclusion was also included to the content of publication.We hope that made corrections will satisfy both the Reviewers and the Editors.

INTRODUCTION:

Line 44: "This study presents the results of the first study of obliteration of GV with vascular coils and 45 injections of CYA glue under the guidance of endoscopic ultrasound (EUS) in Poland".  The authors should change the objective of the study, to something more specific such as " The objective of the study is to report our experience using coils and CYA glue injection in the treatment of gastric varices".

All the necessary changes were introduced into the publication, according to the reviewer’s comments.

METHODS:

The authors should follow the STROBE recommendations for this type of manuscript.

The manuscript has been corrected according to the STROBE recommendation, which contributed to the increase of the essential value of our publication. 

Specify the design of the study. Specify the criteria of inclusion and exclusion of patients.

All the patients included into research fulfilled the follow criteria: conglomerate of gastric varices type GOV 2 and IGV 1 located in the gastric area and varices IGV 2 located in the duodenum with diameter of minimum 5 mm visible in the endosonographic image.

In a case of varices’ diameter ranging from 5 to 10 mm implantation of intravascular coils alone was performed. In a case of varices’ diameter above 10 mm implantation of intravascular coils with cyanoacrylate injections was performed.

Patients with gastric or duodenal varices of diameter under 5mm in endosonographic view were excluded from the study. Moreover, patients under 18 years old and pregnant women were also excluded from the study. Exact patients’ data can be found in the Results section.

We added description of inclusion/exclusion criteria to the text of publication (Method section).

Can the authors include information about why some patients were treated with coils and CYA and other patients only with CYA. Was there a change in the protocol of management of these types of patients?

None of the patients in the study was treated with the cyanoacrylate injections only.  In four cases patients were treated in other medical center with cyanoacrylate injections only, due to bleeding from gastric varices, which happened before our study even began.  Under our study whether implantation of intravascular coils alone was performed or both implantation of intravascular coils together with cyanoacrylate injections.  In a case of varices’ diameter ranging from 5 to 10 mm implantation of intravascular coils alone was performed. In a case of varices’ diameter above 10 mm implantation of intravascular coils with cyanoacrylate injections was performed.

Does the rate of rebleeding was zero during the follow up? In the manuscript was only reported 1 episode of bleeding without known source. In other publications similar to this one the rate of rebleeding was reported around 20% of cases.

During the follow-up no bleeding from the obliterated varices was noticed, apart from one described case which was treated conservatively without endoscopic therapy.

One should stress a fact that in the group described, 10 out of 16 patients experienced bleeding from the varices even before our treatment (6 patients-esophageal varices, 4 patients- gastric varices).

How many patients were under treatment with betablockers?

All the patients included in our study were treated with nonselective betablockers (propranolol) given in maximal tolerated doses.

What was the stage of the disease in case of cirrhotic patients that were included?

9 out of 16 patients in our study were diagnosed gastric varices due to portal thrombosis (Child A- 7/9 patients and Child B -2/9 patients). All the other (7 patients) experienced gastric varices due to liver cirrhosis (Child B- 6/7 patients and Child C- 1/7 patients). As recommended, all those above-mentioned information were covered in detail in our paper.

Were other treatments like TIPS offered to patients included after the treatment with coils.

None of the patients from the study group underwent TIPS procedure. 3 out of 16 patients from the study group who were diagnosed with liver cirrhosis underwent liver transplant.

DISCUSSION

The study " SAFETY AND EFFICACY OF EUS-GUIDED COIL PLUS CYANOACRYLATE VERSUS CONVENTIONAL CYANOACRYLATE TECHNIQUE IN THE TREATMENT OF GASTRIC VARICES: A RANDOMIZED CONTROLLED TRIAL" supports the findings of the study. It would be interesting to include this study in the discussion

Have the authors any explanation for the high prevalence of adverse outcomes reported in the study in comparison to other studies? Was it something related with the experience of the endosonographers in the technique?

Although the prevalence of adverse outcomes is high this is a result of their thorough record and small patients’ number. We want to stress that complications are mild, including mainly pain and fever. Only one patient experienced bleeding from the obliterated varices and was treated conservatively without endoscopic therapy. No other serious complications were observed.   There were no embolic complications. In our opinion complications of that kind are not related to the experience of the endosonographers using the technique.

We hope that now our article will appear more comprehensible and easier to read as well as that the subject was entirely exhausted.

We hope that our corrections will make the manuscript meet the requirements for publication in “Journal of Clinical Medicine”.

With kind regards,

Katarzyna Monika Pawlak

Round 2

Reviewer 1 Report

The paper is improved and the main concerns addressed. The only question is whether all illustrations are original.

Author Response

Thank you very much for your feedback.

All Figures are from studies and are cited accordingly. Figure 1 is drawn by us, while Figure 2 is not, but we can present it ourselves.

Sincerely
Katarzyna Monika Pawlak